# Bacteria and Allergic Diseases

**DOI:** 10.3390/ijms251910298

**Published:** 2024-09-25

**Authors:** Svetlana V. Guryanova

**Affiliations:** 1M.M. Shemyakin and Yu.A. Ovchinnikov Institute of Bioorganic Chemistry RAS, 117997 Moscow, Russia; svgur@ibch.ru; Tel.: +7-(915)3150073; 2Medical Institute, Peoples’ Friendship University of Russia, 117198 Moscow, Russia

**Keywords:** host–microbe interactions, symbiotic bacteria, bacterial bioregulators, allergic inflammation, allergy, food allergy, lung microbiome, asthma, allergic rhinitis

## Abstract

Microorganisms colonize all barrier tissues and are present on the skin and all mucous membranes from birth. Bacteria have many ways of influencing the host organism, including activation of innate immunity receptors by pathogen-associated molecular patterns and synthesis of various chemical compounds, such as vitamins, short-chain fatty acids, bacteriocins, toxins. Bacteria, using extracellular vesicles, can also introduce high-molecular compounds, such as proteins and nucleic acids, into the cell, regulating the metabolic pathways of the host cells. Epithelial cells and immune cells recognize bacterial bioregulators and, depending on the microenvironment and context, determine the direction and intensity of the immune response. A large number of factors influence the maintenance of symbiotic microflora, the diversity of which protects hosts against pathogen colonization. Reduced bacterial diversity is associated with pathogen dominance and allergic diseases of the skin, gastrointestinal tract, and upper and lower respiratory tract, as seen in atopic dermatitis, allergic rhinitis, chronic rhinosinusitis, food allergies, and asthma. Understanding the multifactorial influence of microflora on maintaining health and disease determines the effectiveness of therapy and disease prevention and changes our food preferences and lifestyle to maintain health and active longevity.

## 1. Introduction

Microorganisms colonize skin and mucous membranes from birth and promote the formation of an adequate immune response to pathogenic microorganisms and harmless antigens [1]. Disruption of symbiotic and mutualistic relationships between the microflora and the host organism can lead to various pathological conditions, including allergic diseases [2,3]. It was found that the use of antibiotics during the first 2 years of life was a risk factor for the development of asthma, atopic dermatitis, and allergic rhinitis at the age of five [4,5,6]. This is how the hygiene hypothesis arose, referring to the T-helper type 1/2 (Th1/Th2) model [7,8]. According to the hygiene hypothesis, allergic diseases are associated with insufficient activation of innate immune receptors in early childhood during the development of the immune system. This hypothesis was also confirmed by the discovery that children from urban areas are predominantly affected by allergic diseases compared to children from rural areas [9]. It should be noted that along with the “hygiene hypothesis”, a “counter-regulatory” hypothesis emerged, according to which the epidemic of allergic diseases observed in recent years is a small price to pay for a significant reduction in child mortality, achieved through measures such as improved sanitation, access to drinking water, and vaccination [10]. Microbial colonization of mucosal tissues during infancy may have long-term consequences, such as promoting tolerance to environmental insults or contributing to the development of diseases in later life, including inflammatory bowel disease, allergy, and asthma [11,12]. With increasing understanding of the complexity of the regulation of allergic reactions, other theories have emerged, such as the “biodiversity” hypothesis, which shows a correlation between reduced biodiversity of the skin and mucosal microbiota and the incidence of allergic diseases [13]. In addition, among siblings, younger siblings were found to be less likely to develop allergic diseases [14,15]. With the discovery of bacteria in the meconium of newborns, it became obvious that acquaintance with microorganisms occurs in the womb and the problem of the impact of microflora should be analyzed from periods earlier than birth [16,17,18,19,20].

Modern technologies have revealed associations of genetic polymorphisms in numerous genes involved in the implementation of innate immunity and the formation of tight junctions between epithelial cells with allergic and inflammatory diseases [21,22,23]. However, the discovered polymorphisms could not fully explain the “non-heritability of asthma”, which necessitated determining the influence of environmental factors on epigenetic changes leading to allergic diseases [24]. Among these factors are air pollutants, allergens, and infectious agents in the prenatal and postnatal periods. A modified strategy that includes gene polymorphisms in combination with environmental factors has shown a correlation with allergic diseases [24]. The recently introduced “Epithelial Barrier Theory” includes even more parameters to explain the occurrence of allergic inflammation [25,26]. The Epithelial Barrier Theory combines all previous hypotheses and suggests that toxic substances in hygiene products, as well as microplastics and air pollution, damage the epithelium of our skin, lungs, and gastrointestinal tract [27,28,29,30,31]. In particular, serine proteases and metalloproteinases contained in pollen disrupt the integrity of the lung epithelial barrier by degrading the transmembrane adhesion proteins E-cadherin, claudin-1, and occludin, as well as the cytosolic complex zonula occludens-1 (ZO-1), which leads to an increase in transepithelial permeability [32]. Disruption of tight junctions and an increase in transepithelial permeability facilitate the action of allergens on the epithelial sublayers, promoting sensitization to a wide range of allergens [33].

The increase in allergic diseases since 1960 is also associated with the emergence of 350,000 new toxic compounds that, against the background of genetic susceptibility, contribute to disruption of the epithelial barrier and cause local and systemic inflammatory diseases [31]. Evolutionarily formed relationships between commensal microorganisms and their hosts are under pressure from new, previously unknown chemicals introduced to the immune system, and can significantly modulate immune reactivity along with resident microorganisms. Thus, there is a need to summarize the possible ways in which microorganisms influence the immune system to determine their contribution to the prevention or aggravation of allergic inflammation. The aim of this review was to analyze the influence of bacteria inhabiting the gastrointestinal tract, skin, upper and lower respiratory tract, as well as bioregulators of bacterial origin on allergic diseases.

## 2. Bacteria Modes of Action on Host Cells

Human microflora includes bacteria, archaea, fungi, protozoa, and viruses that inhabit the skin and all mucous membranes [34,35]. Bacterial communities are the most studied and most numerous in the gastrointestinal tract, where their number can reach 10^13^–10^14^ [36]. At the same time, it was found that resident microflora not only help to ferment food [37,38], supply various nutrients and vitamins [17,39,40,41,42], and prevent the colonization of pathogenic microflora [43,44], but also to strengthen the epithelial barrier [45,46]. Such a diverse effect of commensal bacteria on the human body is achieved due to the existence of many ways of activating host cells (Figure 1). First of all, bacteria affect specific receptors of innate immunity located on the surface and in the cytosol of host cells through pathogen-associated molecular patterns (PAMPs). These include peptidoglycans, lipopolysaccharides, flagellin, CpG nucleotides, muramyl peptides, and lectins that activate TLR2, 4, 5, 9, NLR and CLR, respectively [47,48,49,50,51,52]. In this case, lipopolysaccharides and muramyl peptides, which are agonists of TLR and NLR, respectively, may have structural differences characteristic of different strains of microorganisms [53,54,55,56,57]. Activation of PAMP receptors of innate immunity TLR and NLR triggers a cascade of reactions with activation of the transcription factor NFkB, synthesis of proinflammatory cytokines and chemokines, expression of cellular receptors, markers of differentiation, and maturation of immunocompetent cells, which normally leads to elimination of the pathogen [58,59,60]. In this case, cross-interaction occurs between different classes of innate immunity receptors [61,62,63]. In addition, innate immunity receptors cross-interact with other receptors on the cell surface. The microbial metabolite indole-3-propionic acid (IPA) has been shown to suppress enterocyte TNF-α via the pregnane X receptor (PXR). PXR-deficient mice (Nr1i2(-/-)) exhibit a distinctly leaky gut physiology coupled with upregulation of the Toll-like receptor (TLR) signaling pathway. These epithelial barrier defects were corrected in Nr1i2(-/-)Tlr4(-/-) mice [64]. Thus, the bacterial metabolite indole-3-propionic acid, via PXR, normally regulates TLR4 activity. Subsequently, PXR was shown to regulate the intestinal epithelial barrier during inflammation by reducing cytokine-induced myosin light chain kinase expression and c-Jun N-terminal kinase 1/2 activation [65]. The resulting cross-talk and positive and negative feedback loops, depending on the intensity of the stimuli that cause them, ultimately determine the nature of the immune response [52,66].

In addition to specific activity, bacteria have a non-specific effect on host cells by producing amino acids, vitamins, hormones, short-chain fatty acids, bile acid intermediates, bacteriocins, and other chemical compounds [17,39,40,41,42,67,68]. Gut bacterial metabolites exert their biological activity through specific recognition by G protein-coupled receptors (GPRs) [69,70]. GPRs are differentially expressed in different cell types, and the response to the same metabolite can vary depending on the specific roles of the cells [71]. Therefore, the response to gut microbiota-derived metabolites exhibits a huge combinatorial diversity, which poses significant challenges in understanding their effects [72]. Bacteria do not only secrete small molecular compounds; bacterial secretion systems, in particular type III and IV, can deliver proteins and nucleic acids directly into the host cell cytosol, directly influencing cellular metabolic processes including facilitating evasion of immune surveillance [73,74,75].

Another way bacteria influence host cells is through extracellular vesicles, when bacteria can introduce enzymes, RNA, toxins, components of the bacterial membrane, including LPS, into the host cell [76,77,78,79]. In this case, pathogenic bacteria can change intracellular pathways, which allow bacteria to evade the immune response and colonize eukaryotic cells [80,81,82]. Extracellular vesicles of pathogenic bacteria have been studied in more detail than those of commensal bacteria; however, the formation of extracellular vesicles in commensals has been established, with the ability to modulate interaction with the host and immune training [56,82,83,84]. Moreover, it has been established that extracellular vesicles of *Lactobacillus plantarum Q7* have an anti-inflammatory effect [85], and bacterial tryptophan catabolites enclosed in extracellular vesicles can enhance the barrier functions of the intestinal epithelium and regulatory immune responses [86,87].

Interestingly, some biological processes used by cells to combat microorganisms, such as neutrophil extracellular traps (NETs) formation, can be induced by a variety of stimuli, including PAMP activation, as well as independently of PAMPs [88]. NETs formation is a key mechanism of microbe–host cells conversation and is involved in pathogenesis of allergic diseases, especially in asthma exacerbation [89].

Thus, bacteria inhabiting skin and mucous membranes affect all systems and organs in numerous ways, normally contribute to the maintenance of immune homeostasis, and can affect diseases [90,91,92,93].

## 3. Modulation of Allergic Reactions by Bacterial Regulators

Allergic reactions occur in tissues that are adjacent to the external environment and populated by various microorganisms, which, along with heredity and environmental factors, contribute to inflammatory reactions. There are many phenotypes and endotypes of allergic diseases, differing in the causes of occurrence, clinical signs, and immune cells involved in pathological processes [94]. Despite the differences in the manifestation of allergic reactions on the skin, mucous membranes of the gastrointestinal tract, upper and lower respiratory tract, genitals, etc., there are common signs of ensuring the homeostatic state of barrier tissues [95,96,97]. Commensal bacteria play a significant role in maintaining the homeostatic state of barrier tissues, which normally keep innate immune cells active to quickly repel pathogen attacks and ensure tolerance to resident bacteria [1,98,99]. The balance between excessive immune reactivity to harmless antigens and insufficient immune response to dangerous microorganisms can be disrupted by the negative impact of external factors. As a result, the integrity of epithelial barriers can be disrupted, the composition of microflora can be changed and, as a consequence, inflammatory, allergic, and autoimmune reactions can occur [100,101,102].

Depending on the localization, the epithelium of barrier tissues uses different methods of protection from external influences: antimicrobial peptides, mucins, IgA, or various enzymes. However, when recognizing allergens and bacterial metabolites, common response mechanisms can be traced (Figure 2). In the case of recognizing allergens by dendritic antigen-presenting cells (APCs), they have been processed to form complexes of class II major histocompatibility complex (MHC II) molecules and the antigen, migrate to the lymph nodes, and promote the differentiation of naive T cells into Th2 cells [103,104]. At the same time, bacterial bioregulators interacting with APCs through TLRs and NLRs trigger the activation of Th1 and Th17 types by producing cytokines TNF, IL-1β, IL-6, IL-12, IL-23, and IFN γ [105,106,107,108]. When stimulated, naïve T cells secrete IL-2, interferon (IFN)-γ, lymphotoxin, and tumor necrosis factor (TNF)-α, as well as low levels of IL-4, IL-13, and IL-10 [109]. The preferential activation of Th1, Th2, Th17, or Treg depends on the cytokine milieu of the microenvironment [110]. The balance between these endogenously produced cytokines has been determined by the phenotype of the lymphokine-producing primed cells. This balance depends on the genetic background, the nature and strength of the signal, and the activation state of DCs h2 which migrate to the site of inflammation and produce proinflammatory cytokines (IL-4, IL-5, IL-13) to activate eosinophils and stimulate IgE synthesis by B cells and subsequent degranulation of mast cells [111]. Allergen-stimulated epithelial cells secrete IL-25, IL-33, and thymic stromal lymphopoietin (TSLP) to activate innate lymphoid cells type 2 (ILC2), which secrete IL-5, act on eosinophils, and stimulate inflammatory responses [112,113,114]. ILC2 are critical drivers of type 2 (T2) inflammatory responses associated with allergic inflammatory conditions and can secrete IL-4, IL-5, IL-9, and IL-13 [115]. Interestingly, depending on the microenvironment, namely, under the influence of retinoic acid, IL-2, IL-4, IL-10, and IL-33, a subpopulation of KLRG1 + ILC2 may appear which has the ability to produce the anti-inflammatory cytokine IL-10 and exert an anti-inflammatory effect [115]. Cross-interaction of immunocompetent cells, their mutual influence, and sensitivity to the microenvironment normally ensures homeostasis and maintains the integrity of the epithelium [116].

In the case of a compromised epithelial barrier, allergens directly activate APCs and mast cells, enhancing inflammatory responses [117]. In addition, damaged epithelium increases immune cell infiltration, further enhancing dysfunction [117]. When microorganisms and/or their metabolic products affect epithelial cells, the latter produce cytokines (in particular, IL-1, Il-6, and TNF), which activate APCs and T-regulatory cells [118,119,120,121]. Tregs increase immune tolerance during inflammation, highlighting the role of the microbiota in maintaining homeostasis [98]. In addition, Tregs can modulate the responses of other T-cell populations, changing the direction of the immune response [122,123,124,125]. Commensal bacteria are known to induce the production of type I IFN (IFNα and IFNβ) and IL-27 by dendritic cells to influence Tregs [98]. Moreover, IL-27 plays a crucial role in the expression of IL-10 by T-regulatory cells [126,127,128,129]. The importance of IL10 is due to its ability to exert an anti-inflammatory effect, controlling the intensity of immune responses, and maintaining immune homeostasis [115,130,131]. Interleukin 10 is produced by almost all populations of immunocompetent cells—T-helpers, cytotoxic and regulatory T-cells, B-lymphocytes, macrophages, NK-cells, monocytes, dendritic cells, neutrophils, eosinophils and mast cells [132,133,134,135]. At the same time, B-cells secreting IL-10 were isolated into a separate subpopulation, namely regulatory B-cells (Breg), which have anti-inflammatory functions in allergic diseases [136]. The key role of IL-10 in ensuring tolerance is its effect on dendritic cells, which, after interaction with Tregs, promotes allergen-specific and cross-reactive tolerance [137,138]. IL10 is also produced by keratinocytes, intestinal mucosal epithelial cells, and tumor cells [139,140]. The ability of IL10 to suppress the immune response is used by some pathogens to evade immune surveillance [139,140]. In particular, *Helicobacter pylori* stimulate IL10 to survive in the stomach, Mycobacterium tuberculosis induces B cells, and Streptococcus pneumoniae induces neutrophils to produce IL10, which allows the bacteria to colonize their niches [139,141,142,143]. Some viruses, such as HIV, hepatitis C, and hepatitis B, also stimulate IL10 production to suppress immune response and promote viral persistence [140,144]. Importantly, bacteria and their cell wall fragments stimulate the production of both proinflammatory and anti-inflammatory cytokines [11,12,145]. The ability of bacteria to activate anti-inflammatory responses, including IL10 production, limits tissue damage and is important for mediating an effective immune response necessary for host survival [143,146,147].

It should be noted that activation of innate immune receptors in experimental models protects animals from lethal infection with a wide range of pathogens and has therapeutic activity in allergic and oncological diseases [49,148,149]. At the same time, stimulation with commensal bacteria through the induction of type 1 interferons triggers not only antibacterial and antiviral activity, but also the activation of dendritic cells and Treg cells, providing tolerance to commensals [98].

Various combinations of PAMPs can enhance or inhibit the synthesis of cytokines induced by the activation of innate immune receptors, as well as cause activation or suppression of various populations of immunocompetent cells [63]. In particular, PBMCs, monocytes, and dendritic cells secreted high levels of IL-10 when TLR5 was stimulated with flagellin, whereas TLR9 stimulation with CpG oligodeoxynucleotides did not induce IL-10 secretion in any of the three cell types but synergized with flagellin in this induction [63]. Moreover, TLR5 stimulation completely abolished the NK cell cytotoxicity induced by TLR9 stimulation [63].

Hundreds of genes are involved in the regulation of inflammatory processes, including ubiquitinases and deubiquitinases, the activity of which can be modulated by bacterial effectors [76,150]. Bacterial ligases, such as the E3 SopA, NleL, and IpaH families, can act on the principle of molecular mimicry and modulate ubiquitination processes in eukaryotic cells in the direction necessary for bacteria to manipulate the host signaling to facilitate infection [151,152,153,154,155].

## 4. The Impact of Bacteria on Allergic Diseases

Allergic diseases are a serious problem in both developed and developing countries. According to the World Health Organization, 30–40% of the population has one or more allergic diseases [156]. Worldwide, 400 million people suffer from allergic rhinitis, 300 million from asthma, and 250 million people suffer from food allergies [156,157]. The incidence of asthma has increased significantly in recent decades in both developed and developing countries, with more than 40 million new cases diagnosed each year [158]. By 2050, it is predicted that up to 4 billion people worldwide will suffer from asthma, allergic rhinitis, or atopic dermatitis [159]. Allergic diseases impair the quality of life of patients and are a significant burden on healthcare systems. In Europe alone, more than EUR 150 billion are spent annually on combating allergic diseases [160]. With such an increase in allergopathologies, the problem of their treatment and prevention is a pressing issue and requires comprehensive study. The study of bacterial communities in various diseases complements the understanding of the mechanisms of pathology development and is one of the ways to influence the course of the disease.

### 4.1. Gastrointestinal Tract

First contacts of bacteria with the microbiota occur during fetal development in the womb. Live bacteria have been found in amniotic fluid, fetal gut, skin, placenta, and fetal lungs during healthy pregnancy [161,162,163]. Staphylococcus and *Lactobacillus* present in fetal tissues induced the activation of memory T cells in fetal mesenteric lymph nodes [161]. Eighteen taxa were detected in fetal meconium, with *Micrococcaceae* and *Lactobacillus* being the most abundant [163]. Newborns with an asthmatic parent were found to have significantly different meconium bacteria compared to those without an asthmatic parent [164]. The first postnatal stool of neonates with an asthmatic parent was enriched in *Enterobacteriaceae* and *Bacteroidaceae*, depleted in several genera including *Akkermansia*, *Faecalibacterium*, and *Rothia*, was less diverse, and had a significant delay in bacterial richness gain [164]. However, oral administration of *Lactobacillus rhamnosus GG* reversed the deficit in anti-inflammatory lipids, but did not alleviate the deficit in bacterial species diversity [164]. Deficiencies in the bacterial genera *Lachnospira*, *Veillonella*, *Faecalibacterium*, and *Rothia* have been associated with asthma in later life (Table 1) [165]. It turned out that the presence of *Bifidobacterium catenulatum* in the gut microbiota was associated with a higher risk of developing eczema, while *Bifidobacterium breve* and *Bifidobacterium lactis* were identified in healthy children [166,167,168,169,170,171]. *Bifidobacterium* are known to produce short-chain fatty acids (SCFA), and insufficient SCFA content was observed in the intestines of asthma, atopic dermatitis, and allergic rhinitis [172,173,174]. Compensation for the deficiency of *B. breve* alleviated the course of allergic rhinitis and asthma [175]. Another bacterial bioregulator, polysaccharide A, suppresses the production of proinflammatory IL-17 and promotes the expression of IL-10 by CD4 + T cells [176]. Polysaccharide A (PSA) of *Bacteroides fragilis* acts via TLR2 directly on Foxp3(+) regulatory T cells, promoting immune tolerance. PSA-deficient *B. fragilis* is unable to inhibit T helper 17 cell responses and is defective in mucosal colonization [177]. Bacterial bioregulators such as SCFA and polysaccharide A have been implicated in the discovery and mechanism of the anti-inflammatory action of *Bifidobacterium* and *Bacteroides fragilis*. Microbiota-derived metabolites play a major role in the formation of tight junctions of intestinal epithelial cells, which are renewed every 3–5 days. These metabolites influence the expression and function of tight junction-associated proteins and promote a stronger intestinal barrier. Various gut microbiota-derived metabolites such as butyrate, quercetin, indole-3-propionic acid, bile acid intermediates, and L-homoserine can induce increased production of these junctional proteins [64,178,179].

Imbalances in microbiota and their metabolites may affect tight junction function, promoting microbial entry into the bloodstream, and inflammatory diseases. In experimental models, it was shown that, unlike other bifidobacteria, only *B. breve* significantly suppressed airway reactivity to methacholine, reduced acute allergic skin reactions to ovalbumin, and activated IFN-gamma and IL-10 secretion, affecting the maintenance of systemic Th1/Th2 balance [180,181]. In an experimental model, it was also shown that the presence of SCFA in the intestine protected against allergic inflammation in the lungs [182]. In another study, when examining changes in the number of *Lachnospira* (L) and *Clostridium neonatale* (C) bacteria, an increase in the L/C ratio was found in asthmatics [183]. Reduced microbiota diversity is characteristic of various allergic diseases: asthma, allergic rhinitis, and dermatitis [184]. Reduced microbiota diversity characterized by high abundance of the phylum *Firmicutes* and low abundance of *Bacteroidetes,* as well as the presence of the families *Clostridiaceae*, *Ruminococcaceae*, *Lachnospiraceae* or *Erysipelotrichaceae* is associated with hypersensitivity to milk, egg whites, and peanut [185,186]. Decreased biodiversity, consumption of industrially processed foods, food additives, preservatives, and hygiene products increase the allergen load and are considered key factors in the development of allergic diseases [100,187]. For example, cow’s milk can cause allergies, while raw milk has been shown to protect against allergies [187,188].

### 4.2. Upper and Lower Respiratory Tract

Prospective cohort studies have shown that six dominant genera make up the upper respiratory tract microbiota, including *Moraxella*, *Streptococcus*, *Corynebacterium*, *Alloiococcus*, *Haemophilus*, and *Staphylococcus* [162,189]. It has been noted that dysbiosis of the nasal microbiome is characterized by altered bacterial species composition and is associated with the occurrence of infections and subsequent allergic diseases [190]. Asthma patients have been shown to have low levels of Lactobacillus in the nasopharynx [191] and increased numbers of *Bacteroidetes* and *Proteobacteria* [192]. Early colonization of the nasopharynx by *Streptococcus* and *Moraxella* has been associated with asthma [189,191,192,193]. With regard to *Moraxella* species, specific species need to be considered when investigating associations with allergic diseases. For example, *Moraxella catarrhalis* has been reported as a potential risk factor for asthma in children and adults, with neutrophilic inflammation in the lungs observed in adults [194,195,196,197,198].

These findings were confirmed by other studies, which showed that *Proteobacteria*, especially *Haemophilus* spp. and *Moraxella* spp., were significantly increased in neutrophilic asthma, while *Firmicutes*, *Actinobacteria*, and *Saccharibacteria* were reduced in relative abundance compared to healthy controls [199,200]. Bacterial infections with *Haemophilus*, *Moraxella*, or *Streptococcus* spp. induce the secretion of IL-17, which in turn attracts neutrophils to the airways [201]. Neutrophilic asthma is characterized by an impaired response to corticosteroid treatment. The presence of bacteria in sputum has led to the development of antibiotic treatments for neutrophilic asthma, particularly azithromycin [201]. Eosinophilic asthma has been correlated with the presence of *Tropheryma whipplei* in sputum [200,202].

*Klebsiella* has been detected in bronchial lavage samples in asthmatics [203]. A study of bronchial lavage in children and adults revealed a predominance of *Proteobacteria*, particularly *Haemophilus* spp., in asthmatics, while *Bacteroidetes*, particularly *Prevotella* spp., were less common than in healthy controls [204].

Low levels of *Bacteroidetes* are present in the upper airways of patients with chronic rhinosinusitis [205]. Nasal lavage fluid samples from patients with chronic rhinosinusitis had lower bacterial diversity, and higher bacterial abundance of Staphylococcus aureus and its extracellular vesicles [205].

A study by Ruokolainen L. and colleagues examined the dependence of allergic rhinitis on genetic factors and lifestyle. It turned out that allergic manifestations and sensitization to common allergens were significantly influenced by the microbiota of the nasal epithelium and skin and determined by lifestyle. *Acinetobacter*, in particular, was associated with the absence of allergic rhinitis [206].

The study of bacterial communities in the upper and lower respiratory tract serves as a basis for the development of asthma therapy using probiotics [207]. However, it should be taken into account that the formation of the lung microbiome is associated with aerobic bacteria, and the microbiota of the gastrointestinal tract with anaerobic bacteria [208,209]. In addition, the effect of probiotics may depend on diet, and in some cases aggravate allergic inflammation. For example, the popular probiotic Akkermansia, containing the bacterium *Akkermansia muciniphila*, aggravates food allergies in the absence of fiber by activating the Th2 pathway [210]. The use of modern methods, including bioinformatics analysis and machine learning, can help in predicting possible reactions to bacteria and using bacteria as diagnostic markers of diseases [211]. Intensive research on bacterial metabolites has been conducted in the intestine. The study of bacterial bioregulators of microorganisms inhabiting the upper and lower respiratory tract is associated with the difficulty of excluding bacterial contamination from the oral cavity; however, their identification can reveal new molecules of therapeutic value.

Bioregulators of bacterial origin—PAMPs and bacterial metabolites—play a central role in regulating host homeostasis [45,46,47,72]. It is believed that bacterial metabolites and their derivatives can be effective treatments [212].

### 4.3. Skin

Microorganisms colonize the skin immediately after birth. The composition of the skin microbiota of a newborn during the first three months depends on the method of delivery. Infants born vaginally acquired bacterial communities similar to their mother’s vaginal microbiota, dominated by *Lactobacillus*, *Prevotella*, or *Sneathia* spp., whereas infants born by cesarean section harbored bacterial communities similar to those found on the skin surface, dominated by *Staphylococcus*, *Corynebacterium*, and *Propionibacterium* spp. [213]. The neonatal microbiota was relatively uniform across all body sites at delivery, with the exception of neonatal meconium. However, by 6 weeks, the infant microbiota had expanded significantly, become more diverse, and was not affected by delivery mode [214]. However, as early as 3 months, infants can be diagnosed with atopic dermatitis (AD) if their skin is colonized by *Staphylococcus aureus* [215,216,217]. In pediatric AD, the skin commensal *S. epidermidis* was significantly reduced in abundance, and *Streptococcus*, *Propionibacterium*, and *Corynebacterium* species were significantly reduced [216]. Skin commensals *S. epidermidis* and *Staphylococcus hominis* protect the host from pathogens, including *Staphylococcus aureus*, which stimulate the host to produce antimicrobial peptides such as cathelicidin and human beta-defensin (HBD), and directly produce PSM-γ and PSM-δ, which inhibit the growth of pathogenic bacteria [218]. It should be noted that local skin anatomy, lipid content, and pH also influence the occurrence of AD [219]. The skin is known to have an acidic pH, which promotes optimal barrier function. In humans, the skin surface pH is close to neutral at birth (pH 6.5), and it takes several weeks after birth for the pH to reach the normal range of 5.4–5.9 [220]. *Staphylococcus aureus* can exist in a wide pH range, tolerate acid loads, and form biofilms [221,222]. *Staphylococcus aureus* exerts its pathogenic effects through the production of virulence factors such as α-toxin, protein A (SpA), lipotechoic acid (LTA), phenol-soluble modulin (PSM)-α, and proteases that can damage keratinocytes [223,224,225].

Skin colonization by *S. aureus* has been associated with food allergies. In a study of 718 children aged 0–18 years and 640 infants with severe eczema, an association was found between *S. aureus* colonization and egg or peanut allergy [226,227,228]. The increased risk of peanut and egg allergy in the first 5 years of life was independent of AD severity [227,228]. *S. aureus* enterotoxins are classified as superantigens (SAgs) due to their ability to activate B and T cells and induce immunoglobulin E production against SAgs in individuals with atopic dermatitis, allergic rhinitis, chronic sinusitis, and asthma [229,230,231,232,233]. Severe atopic dermatitis has been found to be characterized by an expansion of circulating Th2 and Th22 cells, but not Th17, in the skin-homing T cell population [233,234]. It should be noted that the virulence of *S. aureus* can be altered by the presence of other bacteria. In particular, the presence of *Corynebacterium* promotes the transition of *S. aureus* from a virulent to a commensal state, which explains the presence of *S. aureus* on the skin and in the nasal cavity in 25% of healthy individuals [235].

An association between *Acinetobacter* skin colonization and allergic rhinitis has been documented. Thus, when identifying the causes of allergic rhinitis in populations with a similar genotype but different lifestyle, genome-wide association analysis revealed that lifestyle and environment affect gene expression in blood mononuclear cells and depend on the microbiome of the skin and nasal epithelium [206]. Allergic rhinitis was found to be correlated with increased activity in 261 genes of the innate immune pathway, with statistically significant differences observed with decreased *Acinetobacter* representation on the skin and in the nasal cavity. In addition, a high number of expressed genes leads to a more balanced innate immunity and is associated with a low prevalence of allergies [206].

**Table 1 ijms-25-10298-t001:** Bacteria in allergic diseases.

	Bacteria or Substances	Disease	Reference
Gastrointestinal tract	Deficiency of *Lachnospira*, *Veillonella*, *Rothia* and *Faecalibacterium*	Asthma	[165]
	An increase in the *Lachnospira/Clostridium neonatale* ratio		[183]
	Deficiency of odd-chain fatty acids, 10-nonadecenoate and 10-hepatadecenoate Enriched for the aconitate		[164]
	Deficiency of short-chain fatty acids		[172,182]
	Presence of *Bifidobacterium catenulatum*	Atopic dermatitis	[166,167,168]
	Deficiency of *Bifidobacterium breve*		[169]
	An increase of the species *Firmicutes* such as *Clostridium* and deficiency of *Bacteroides*		[170]
	Deficiency of oligosaccharides and short-chain fatty acids		[172]
	Deficiency of *Bifidobacterium breve*	Allergic rhinitis	[175]
	Deficiency of *Bifidobacterium lactis*		[171]
	Deficiency of short-chain fatty acids		[173]
	Presence of *Enterobacter* and deficiency of *Bifidobacterium*	Chronic rhinosinusitis	[174]
	Presence of *Clostridiaceae*, *Ruminococcaceae*, *Lachnospiraceae* and *Erysipelotrichaceae*	Food allergy	[185]
	Deficiency of *Lactobacillales*, *Bacteroidales*		[186]
Upper and lower respiratory tract	*Streptococcus* in nasopharynx	Asthma	[189]
	Predominance of *Moraxella* in nasopharynx		[193,194,195]
	Low abundance of *Lactobacillus* in nasopharynx		[191]
	Low abundance of *Firmicutes*, *Actinobacteria* and *Saccharibacteria* in nasopharynx		[199]
	*Haemophilus influenza*, *Moraxella catarrhalis* and *Tropheryma whipplei* in sputum		[200,202]
	*Moraxella catarrhalis*, *Haemophilus* and *Streptococcus* in sputum		[198]
	Enriched with taxa from *Bacteroidetes* and *Proteobacteria* in nasopharynx		[192]
	*Klebsiella* in bronchial swab samples		[203]
	Low abundance of *Bacteroidetes* and predominance of *Staphylococcus aureus* in nasal lavage	Chronic rhinosinusitis	[205]
	Low abundance of *Acinetobacter*	Allergic rhinitis	[206]
Skin	Predominance of *Staphylococcus aureus*	Atopic dermatitis	[215,216,217]
	Deficiency of *Staphylococcus epidermidis*, *Streptococcus*, *Propionibacterium* and *Corynebacterium*		[216]
	Predominance of *Staphylococcus aureus* and deficiency of *Staphylococcus epidermidis* and *Staphylococcus hominis*		[218]
	Predominance of *Staphylococcus aureus*	Food allergy	[226,227,228]
	Low abundance of *Acinetobacter*	Allergic rhinitis	[206]

## 5. Concluding Remarks

New approaches to the diagnosis and therapy of asthma and associated diseases are being developed, including metagenomic studies, transcriptomics, systems biology, and cell therapy [150,236,237,238,239,240]. At the same time, it is necessary to take into account a large number of factors influencing the occurrence of allergic diseases, in particular, the influence of the mother’s microbiome, her diet, and the use of antibiotics during pregnancy on the formation of the allergic phenotype of her future child [162,241]. In addition, it is necessary to take into account the influence of viruses, fungi, and archaea that are part of the human microbiome and enter into complex mutualistic, symbiotic, and antagonistic interactions. It is important to determine not only the phyla and genera of bacteria associated with diseases, but also to study the species that can compete with each other for the habitat, as is shown by the example of *S. epidermidis* and *S. aureus* (phyla *Firmicutes*, genera *Staphylococci*).

The accumulated knowledge about the features of the impact of various microorganisms on humans makes it possible to prevent allergic diseases using both whole bacteria and their fragments and metabolites [162,242,243,244]. Currently, active research is underway to study the effect of microorganisms or their fragments on their hosts, while determining the effect of microorganism metabolites on humans requires more complex technologies. Modern methods developed for registering low-molecular compounds, as well as modern technologies, including machine learning, inspire hope for significant achievements in the near future. The ability of microbiota and bacterial metabolites to strengthen the integrity of the intestinal barrier, provide protection against inflammation, improve nutrient uptake pathways, and improve protection against age-related diseases has been associated with increased lifespan and health [245,246].

Understanding the mechanisms by which the immune response to various microorganisms and the substances they produce can help to discover how our immune system can detect commensal and pathogenic bacteria and develop new strategies to combat pathogens and allergic diseases. The numerous ways in which microorganisms affect their hosts require the consideration of many thousands of signaling pathways to identify associations with diseases. It is clear that maintaining health requires the participation of a large number of factors, provided by biodiversity. A thorough analysis of the influence of all possible bioregulators of bacterial origin and their cross-interaction, taking into account specific genetic characteristics and environmental factors, will allow us to accurately assess the events taking place and develop personalized therapies and strategies for the prevention of allergic diseases. Our knowledge of the influence of microflora on the functioning of all systems and organs leads to changes our food preferences and lifestyle.

## Figures and Tables

**Figure 1 ijms-25-10298-f001:**
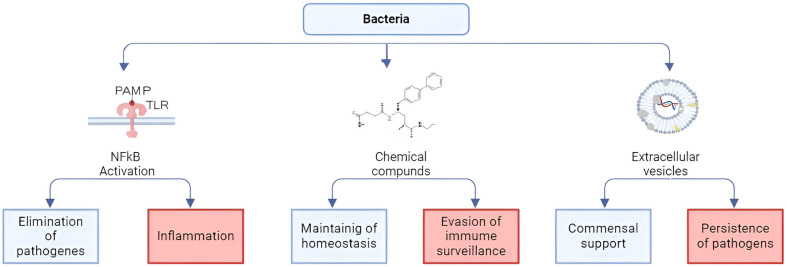
Bacteria modes of action on host cells. Bacteria activate host cells via different ways. First of all, bacteria affect specific receptors of the innate immunity located on the surface and in the cytosol of host cells through pathogen-associated molecular patterns (PAMPs). Activation of PAMPs receptors of innate immunity TLR and NLR triggers a cascade of reactions with activation of the transcription factor NFkB, which normally leads to pathogen elimination. When the NFkB pathway is aberrantly active, inflammation occurs. The second way of action is a non-specific penetration through the host cells of amino acids, vitamins, hormones, short-chain fatty acids, bile acid intermediates, bacteriocins, and other chemical compounds produced by bacteria. Bacterial secretion systems can deliver proteins and nucleic acids directly into the host cell cytosol, directly influencing cellular metabolic processes including facilitating evasion of immune surveillance. The third way bacteria influence host cells is through extracellular vesicles. Extracellular vesicles from commensal bacteria help maintain trained immunity, whereas extracellular vesicles from pathogenic bacteria can alter intracellular pathways and colonize eukaryotic cells.

**Figure 2 ijms-25-10298-f002:**
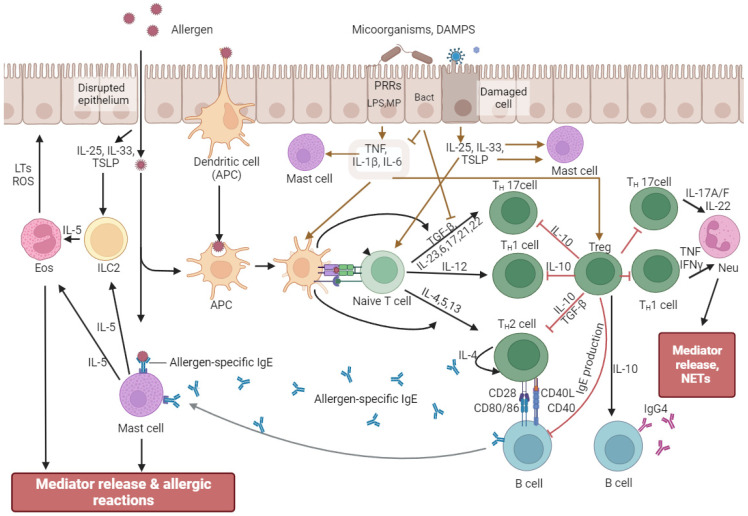
Mechanism of formation and attenuation of allergic reactions. Allergens are recognized and processed by dendritic antigen-presenting cells (APCs) to form complexes of major histocompatibility complex class II (MHC II) molecules and antigens, which promote differentiation of naive T cells into Th2 cells. Th2 cells produce proinflammatory cytokines (IL-4, IL-5, IL-13) to activate eosinophils and stimulate IgE synthesis by B cells and subsequent degranulation of mast cells. When microorganisms and/or their metabolic products (brown arrows) affect epithelial cells, the latter produce cytokines (in particular, IL-1, Il-6, TNF), which activate APCs and T regulatory cells (Treg). Tregs can modulate the responses of other T cell populations, changing the direction of the immune response. Allergen-stimulated epithelial cells secrete IL-25, IL-33, and thymic stromal lymphopoietin to activate ILC2, which secretes IL-5, which affects eosinophils, stimulating inflammatory responses. When the epithelial barrier is compromised, allergens directly activate APCs and mast cells, enhancing inflammatory responses. Abbreviations: APCs—antigen-presenting cells; DAMPs—danger-associated molecular patterns; Neu—neutrophils; Eos—eosinophils; ILC2—group 2 innate lymphoid cells; NETs—neutrophil extracellular traps; LTs—leukotrienes; ROS—reactive oxygen species; TNF—tumor necrosis factor; TSLP—thymic stromal lymphopoietin.

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
