# Peer review of "Bacteria and Allergic Diseases"

_ijms, 2024, doi:10.3390/ijms251910298_

Round 1

Reviewer 1 Report

Comments and Suggestions for Authors

In a review article (Bacteria and allergic diseases), the author summarizes the effects of bacterial dysbiosis in various organ systems (skin, gastrointestinal tract, upper and lower respiratory tract) in patients with various allergic diseases, such as asthma, allergic rhinitis, allergic rhinosinusitis and food allergy, based on the scientific knowledge published so far. Allergic diseases and bacteria are not specifically covered throughout the chapters, but are mentioned in the chapters related to organ systems. I think that the work would be more transparent if the connection between bacteria (or rather bacterial dysbiosis) and allergic diseases were presented in this way.

Author Response

Dear Reviewer, Thank you for reading the manuscript and for your valuable suggestions. Indeed, it was possible to start from diseases rather than organ systems and show their connection with bacteria. We chose an approach based on the analysis of allergic diseases of organs in relation to common bacterial communities inhabiting organs and causing different allergic diseases. In addition, we also preliminarily considered the possibility of starting from different bacterial species and showing which diseases they cause. We chose an approach based on the fact that bacterial communities inhabiting one organ system and causing different allergic diseases can provide important information for the purpose of systematization and identification of common modes of action.

Reviewer 2 Report

Comments and Suggestions for Authors

I carefully read the article by Svetlana V. Guryanova. and find the topic very interesting.

I would like to address a few suggestions to authors that may improve the manuscript

Abstract

Page 1. Line 9

Please add explanation for PAMPs

1.      Introduction

Introduction of this manuscript is too long and must be shortened. Fragments with no new information, practically nearly all text up to "The aim of this review..." should be re-written.

2. Bacteria modes of action on host cells

In this section, must be mentioned the mechanism of NETs, given that it is one of the key mechanisms of microbe-host cells conversation, that involvement in pathogenesis of allergic diseases, especially, in asthma exacerbation.

Page 4, line 143

Please write the microorganisms in italics "Lactobacillus plantarum"

Figure 1. Bacteria modes of action on host cells. The figure 1 is non informative and should be changed. Moreover, for all figures the legends must be written.

Page 10, line 406

Please write the microorganisms in italics “S. aureus”

Author Response

Response to Reviewer’s comments

Dear  Reviewer,

I sincerely thank you for comments which helped to improve the quality of the manuscript ijms-2169749. The manuscript was revised point by point according to your kind advices and suggestions. I sincerely hope that this version of the manuscript will make a contribution to the International Journal of Molecular Sciences. All of the corrected parts were marked up.

Response to Reviewer’s comments

Comment 1: Abstract, Page 1. Line 9: Please add explanation for PAMPs

Response: The explanation for PAMPs was added.

Comment 2: Introduction of this manuscript is too long and must be shortened. Fragments with no new information, practically nearly all text up to "The aim of this review..." should be re-written.

Response: Introduction of this manuscript was shortened.

Comment 3: 2. Bacteria modes of action on host cells

In this section, must be mentioned the mechanism of NETs, given that it is one of the key mechanisms of microbe-host cells conversation, that involvement in pathogenesis of allergic diseases, especially, in asthma exacerbation.

Response: The information about NETs formation and its influence on pathogenesis of allergic diseases, especially, in asthma exacerbation was added.

Comment 4: Page 4, line 143: Please write the microorganisms in italics "Lactobacillus plantarum"

Response: “Lactobacillus plantarum” was written in italics.

Comment 5: Figure 1. Bacteria modes of action on host cells. The figure 1 is non informative and should be changed. Moreover, for all figures the legends must be written.

Response: The legend to Figure 1 was added.

Comment 5: Page 10, line 406: Please write the microorganisms in italics “S. aureus”

Response: S. aureus” was written in italics.
